# *ASIP*, *AHCY* and *ITCH* Genes Are Associated with the Coat Color of Local Goats (*Capra hircus*) of Southwestern China

**DOI:** 10.3390/ani15131849

**Published:** 2025-06-23

**Authors:** Linyun Zhang, Shengnan Zhao, Houmo Yu, Yixin Duan, Jipan Zhang, Naiyi Xu, Yongju Zhao

**Affiliations:** College of Animal Science and Technology, Southwest University, Chongqing Key Laboratory of Herbivore Science, Chongqing Key Laboratory of Forage & Herbivore, Chongqing Engineering Research Center for Herbivores Resource Protection and Utilization, Chongqing Herbivore Engineering Research Center, Chongqing 400715, China; lyzhang023@163.com (L.Z.); zsnyolo@163.com (S.Z.); yhm7791@163.com (H.Y.); duanyx315@163.com (Y.D.); jpanzhang@live.com (J.Z.)

**Keywords:** goat, coat color, GWAS, haplotype

## Abstract

Coat color is an important economic trait in goats, influencing breed identification and market value. Chongqing goats, representative local breeds in Southwestern China, exhibit diverse coat colors; however, the genetic basis underlying this variation remains poorly understood. In this study, we identified genes associated with the coat color of Chongqing goats via a genome-wide association study (GWAS). The results revealed that the *ASIP*, *AHCY,* and *ITCH* genes on chromosome 13 play a significant role in coat color variation in Chongqing goats. Further haplotype and functional analyses supported the association of these genes with coat color, with *ASIP* and *AHCY* related to black coat color and the *ITCH* gene potentially influencing white coat color expression. These findings enhance our understanding of the genetic mechanisms regulating coat color in goats and provide valuable genetic resources for breed conservation and improvement.

## 1. Introduction

Goats, as highly adaptable and economically sustainable livestock, exhibit coat coloration that not only determines the market value of their pelts and fibers but also correlates with multiple significant traits, including growth performance, disease resistance, and environmental adaptability. Additionally, coat color serves as a critical identifier for individual recognition, breed (strain) classification, and taxonomic differentiation. The expression and regulation of coat coloration represent a genetically intricate process, governed by the polygenic interactions of multiple loci that collectively orchestrate pigment synthesis, distribution, and phenotypic manifestation [1]. The diverse coat color observed in goats represents a fascinating biological phenomenon. This diversity is closely linked to significant variations in melanin content among different breeds, which is reflected in the wide spectrum of pigmentation patterns across goat populations [2]. In recent years, substantial progress has been made in elucidating the genetic mechanisms responsible for coat color variation in goats, largely driven by advancements in high-throughput sequencing technologies. Key genes such as *ASIP* and *MC1R* have been functionally characterized, with their regulatory mechanisms becoming increasingly well understood [3,4,5]. These genes play pivotal roles in modulating the biosynthesis of eumelanin and pheomelanin, thereby governing both the intensity and spatial distribution of coat pigmentation patterns.

China possesses abundant indigenous goat genetic resources. Chongqing goats represent typical local breeds in Southwestern China, with six local varieties recognized as national-level genetic resources: Dazu black goat (DZ), Yudong black goat (YD), Banjiao goat (BJ), Hechuan white goat (HC), Chuandong white goat (CD), and Youzhou black goat (YZ). Due to long-term adaptation to unique natural environment and traditional farming practices, these goats have developed distinct regional coat color patterns. In addition to the black and white coat colors commonly found in these breeds, YZ exhibits a relatively unique phenotype characterized by a predominantly white coat with a black mid-dorsal stripe. The distinct phenotypic characteristics of all six local breeds of Chongqing are illustrated in Appendix A. Research on the genetic mechanisms underlying coat color in Chongqing indigenous goats remains relatively limited, and the regulatory networks controlling coat color in these breeds have yet to be fully elucidated. With the advancement of genomic technologies, GWASs have become a powerful tool for identifying genetic variations influencing these traits [6]. Compared to genotyping arrays, whole-genome resequencing provides a significantly larger amount of variant information, enabling more precise analysis results [7]. Therefore, in this study, we used whole-genome resequencing data that our lab previously generated to perform a GWAS.

Given the crucial role of coat color in breed improvement and the conservation of local genetic resources, this study aims to identify candidate genomic regions associated with coat color in indigenous Chongqing goats using whole-genome resequencing. This study of Chongqing goats with different coat colors utilizes a GWAS to identify candidate functional genes related to coat color. In addition, haplotype analysis, candidate gene protein interactions, functional annotation, and pathway enrichment analysis are performed to further explore the potential regulatory roles of these genes in coat color determination.

## 2. Materials and Methods

A complete list of abbreviations used in this manuscript is provided in the abbreviations section Appendix A.

### 2.1. Data and Quality Control

We used 776 goat whole-genome resequencing data determined by our previously generated data (NCBI accession numbers: PRJNA1212315) [8] using the DNBSEQ-T7 platform and aligned the clean reads to the goat reference genome (ARS1.2) with BWA. Quality control was performed with PLINK [9], removing SNPs with missing rates > 10%, minor allele frequency < 1%, and Hardy–Weinberg equilibrium *p*-values < 1 × 10^−6^.

### 2.2. Genome-Wide Association Study

In this study, a GWAS between single-nucleotide polymorphisms (SNPs) and coat color was conducted using a mixed linear model (MLM) in the GEMMA [10] software. The model is specified as follows:(1)y=Xβ+Zg+Qk+e

In this equation, *y* represents the phenotypic data, *β* is the vector of fixed effects excluding SNPs and kinship (such as PCA, sex, etc.), *g* is the vector of SNP effects, *k* represents the vector of random effects (kinship), and *e* is the vector of residual effects. *X*, *Z*, and *Q* are the corresponding design matrices.

The GWAS results were visualized using a Manhattan plot and QQ plot generated with the “qqman” [11] and “CMplot” packages [12] in R.

To determine the significance thresholds, Bonferroni correction was applied. The genome-wide significance threshold was set at 1/n, while an extremely significant threshold was established at 0.05/n: genes existing within ±200 kb of the SNPs noted.

### 2.3. Haplotype Analysis

The haplotype analysis was explored using Beagle [13] and the “pegas” [14] and “ape” [15] packages in R; the network diagram was drawn using Network 10.02 [16].

### 2.4. Gene Function Annotation and Pathway Analysis

The protein–protein interaction (PPI) network diagram was constructed by using the STRING database [17]. The gene ontology (GO) database classifies gene functions into three main categories: cellular component (CC), molecular function (MF), and biological process (BP). Kyoto Encyclopedia of Genes and Genomes (KEGG) is a database for metabolic pathways. Candidate genes were identified based on their positions in the ARS1.2, using the upstream and downstream 200 kb distances of significant loci as the interval. SNP loci were annotated to corresponding genes using bedtools software version 2.18. GO and KEGG functional enrichment analyses were performed using the DAVID online tool (https://davidbioinformatics.nih.gov/ accessed on 20 January 2025), and the results were visualized using the “ggplot2” package [18] in R.

## 3. Results

### 3.1. GWAS of Coat Color in Chongqing Local Goats

A GWAS was performed for the six local goat breeds from Chongqing, encompassing three coat color types: black (DZ, YD), white (BJ, HC, CD), and mainly white with a black mid-dorsal stripe (YZ), abbreviated as black-white. The results revealed a highly significant region on chromosome 13 associated with the *ASIP* gene, as well as significant regions on chromosomes 4 and 26 associated with the *TPK1* and *PCDH15* genes, respectively, and the *KIT* gene was identified within the regulatory region spanning ±200 kb around a highly significant SNP on chromosome 6 (Figure 1, Appendix A).

In addition to black and white coat colors, Chongqing indigenous goats also include the black-white YZ breed. To investigate the genetic basis of coat color variation, we conducted a three-way comparison among these coat color groups, as well as pairwise comparisons between the two groups in combination. This strategy aimed to identify genetic variants primarily responsible for black or white coat color formation and to determine whether a unique genetic mechanism underlies the development of black-white patterned coats. First, we performed a GWAS by comparing the black coat color group (DZ, YD) with white coat color group (BJ, HC, CD). The significant SNPs on chromosome 13 were mapped to *ASIP* and *AHCY*. Additionally, within the ±200 kb region surrounding the significant SNPs on chromosome 13, several genes were identified, including *ITCH*, *EIF2S2*, *RALY*, and *LOC10219053* and several genes which were mapped on chromosomes 2, 10, 15, and 18, such as *PDE1A*, *RYR3* and *CMIP*, were also detected (Figure 2A, Appendix A).

Goats with a black coat color (DZ, YD) were then compared with those with a black-white coat color (YZ), revealing a highly significant SNP region on chromosome 13. The significant SNPs were mapped to the *ASIP*, *BPIFB4*, and *DNMT3B* genes, while the ±200 kb flanking region contained *AHCY*, *ITCH*, and *EIF2S2*, as well as significant regions on chromosomes 2, 5, 7, 8, 9, 10, 14, 17, 20, and 27 associated with genes such as *IRS1*, *DNM2,* and *ESR1,* which may be related to the expression of coat color in goats (Figure 2B, Appendix A). Finally, comparing goats with a white coat color (BJ, HC, CD) with those with a black-white coat color (YZ) revealed the most significant SNP region located on chromosome 13. The significant SNPs were mapped to the *ASIP* and *ITCH*, with the ±200 kb flanking region containing genes such as *AHCY*, *EIF2S2*, *RALY*, and *LOC102190531.* In addition, significant regions on chromosomes 1, 4, 11, 16, 25, and 26, containing genes such as *CTNNA2*, *SLC29A4*, and *LOC102185739*, were also detected and may be associated with coat color variation in goats (Figure 2C, Appendix A).

### 3.2. Haplotype Analysis of Coat Color Significant SNP Region

Haplotypes were constructed using the significant regions identified through the GWAS across all the above four combinations (Figure 1 and Figure 2) to further validate the association between candidate genes and coat color variation. The *ASIP* gene exhibited strong associations across multiple statistical models, and a highly significant region within the *ASIP* gene (chr13: 63,232,000–63,263,000 bp) was selected for haplotype construction (Figure 3A). The results reveal that individuals in the black coat color group (red) possessed distinct haplotypes within this region, which were closely related to each other and clearly differentiated from those observed in the white coat color (blue) and black-white coat color (purple) groups.

In the analysis comparing black with white and black-white coat colors, significant SNPs were consistently identified within or near the *AHCY* gene across groups. Therefore, haplotypes were constructed, and a haplotype network was generated for the highly significant region of the *AHCY* gene (chr13: 63,257,000–63,276,000 bp) (Figure 3B). Within this region, the black coat color group (red) possessed a unique haplotype, with most individuals and haplotypes clustering closely together in the network, suggesting a strong association with this pigmentation phenotype. Additionally, the white coat color group (blue) and the black-white coat color group (purple) exhibited distinct but less tightly clustered haplotypes in this region.

In the analysis comparing white with black-white coat colors, significant SNPs were identified within the *ITCH* gene. Additionally, *ITCH* was also found to be near significant SNPs in the comparisons between black and white coat color groups, as well as between black coat color and black-white coat color groups. Haplotypes were constructed, and a haplotype network was generated for the *ITCH* gene (chr13: 63,363,104–63,462,630) (Figure 3C). In the haplotypes of the *ITCH* gene, the black coat color group (red) possessed unique haplotypes, though most individuals showed relatively minor differences from other coat colors. However, the white coat color group (blue) exhibited relatively distinct and clustered haplotypes, suggesting that the *ITCH* gene may influence coat color by affecting the expression of genes associated with white coat color in goats.

### 3.3. Results of Gene Function Annotation and Pathway Analysis

To investigate the functional relevance of the genes prioritized in coat color regulation, the PPI network was constructed using significant genes (*ASIP*, *KIT*, *AHCY*, *ITCH*, and *RALY*) identified via the GWAS (Figure 4A). Notably, *ASIP* and *KIT* occupy hub positions within the network topology, exhibiting direct or indirect interactions with core regulators of melanogenesis and pigment deposition, including *MC1R*, *TYR*, and *TYRP1*. This connectivity pattern underscores their pivotal regulatory roles in modulating coat color phenotypes. Furthermore, *AHCY*, *ITCH*, and *RALY* demonstrate network connectivity with *ASIP* or *MC1R*, suggesting their potential involvement in melanogenic signaling cascades via indirect regulatory mechanisms.

KEGG pathway enrichment was performed on the differentially enriched genes of coat color in Chongqing local goats (Figure 4B). Among the identified pathways, several pathways are potentially involved in pigmentation regulation. Notably, the PI3K-Akt signaling pathway and pathways in cancer were significantly enriched, both of which are known to regulate cell proliferation, differentiation, and processes key to survival in melanocyte development and function [19]. Additionally, pathways such as salmonella infection and leukocyte transendothelial migration may suggest a potential link between immune regulation and coat color determination [20]. These findings imply that coat color variation may be influenced by multiple signaling pathways, particularly those involved in melanogenesis, cellular signaling, and immune-related processes.

The GO enrichment analysis identified functional categories associated with genes potentially involved in coat color variation. The GO terms are classified into three categories (Figure 4C): biological processes (GO_BP, red), cellular components (GO_CC, blue), and molecular functions (GO_MF, green). The most enriched biological processes included protein phosphorylation and intracellular signal transduction, both of which are closely associated with melanogenesis and pigment cell signaling [21]. Additionally, the enrichment of GO terms related to receptor activity, lipid binding, and kinase signaling pathways suggests a potential role in regulating pigment cell differentiation and melanin synthesis [22]. These functional categories collectively indicate that the genes associated with coat color may influence pigmentation in local Chongqing goats through mechanisms involving signal transduction, phosphorylation, and metabolic regulation.

## 4. Discussion

Coat color is a critical economic trait in goats, and recent advances in molecular biology have significantly enhanced our understanding of the genetic basis underlying this trait. Our study identified key genetic regions associated with coat color in local goat breeds from Chongqing, including the genes *ASIP*, *AHCY*, and *ITCH*. *ASIP*, located on chromosome 13, plays a crucial role in pigmentation regulation across various species, including goats, with studies highlighting its key involvement in coat color patterns [3]. During the association analysis of the three different coat colors, a significant regulatory SNP region on chromosome 6 was mapped to the *KIT* gene, which is significantly associated with coat color. We identified the genes (*AHCY*, *ITCH*, and *KIT*) related to coat color, which were also observed in breeds such as the Angora, Chongming white goat, cashmere goat, Pakistani Angora, and Barbari goats [23,24,25,26,27]. However, many investigations still rely on SNP arrays due to their low cost and high throughput. And SNP chips are limited to predefined loci and may fail to capture rare or novel variants. Compared to previous research, our study employed whole-genome resequencing, which offers broader genomic coverage and higher resolution, enabling more comprehensive variant detection. Additionally, many previous studies have been limited by small sample sizes, which reduce statistical power for detecting minor allele effects and increase the risk of false negatives. In this study, we utilized a large dataset comprising 776 individuals, thereby enhancing the robustness and reliability of association signals related to coat color in goats.

In this study, we found that the *AHCY* and *ITCH* genes were significantly associated with coat color variation. In the *AHCY* gene haplotypes, the black coat color group was almost entirely distinct from other coat color groups; the haplotype distribution in different groups of *AHCY* is similar to that of the *ASIP* gene, suggesting that the expression in this region may affect the black coat color of goats. In the *ITCH* gene haplotypes, two independent regions were observed in the white coat color group, one of which was almost entirely found in BJ, while the other showed a high frequency in HC, indicating potential breed-specific differences in white coat expression. In particular, there were multiple highly significant SNPs of *ITCH* in the comparison between white coat color and black-white coat color goats. These SNPs showed significant differences from the other groups. Combined with its specificity regarding the white population in haplotypes, it may have a certain relationship with the degree of expression of the white coat phenotype in goats. More importantly, the association between the *ITCH* gene and white coat color in goats has not been previously explored, so the current study provides novel insights into the genetic basis of white coat color expression in goats. In addition, several significant SNPs were identified within the *TPK1* gene on chromosome 1, which has not been previously linked to pigmentation. This finding suggests that *TPK1* is a potential novel candidate gene involved in coat color regulation in goats. In addition, multiple genes located within 200 kb upstream and downstream of significant SNPs were detected, indicating that a broader regulatory network may contribute to coat color variation. These genes (e.g., *TPK1*, *EIF2S2*, and *LOC102190531*), which have hardly been reported, provide valuable targets for future studies on the genetic architecture and molecular mechanisms underlying pigmentation traits in goats. Further research and verification can be conducted in the future. Nevertheless, all significant SNPs located within the ASIP gene region were annotated as intron variants or 5′ untranslated region (5′UTR) variants (Appendix A). None of them resulted in protein-coding changes, suggesting that these loci may influence coat color through transcriptional regulation or splicing-related mechanisms rather than direct amino acid alterations. The precise regulatory roles of these intronic and UTR variants remain to be elucidated and warrant further functional investigations. Our study fills the gap in research on the coat color of Chongqing goats and provides a comprehensive understanding of goat coat color. These findings may be applied to the genetic improvement of local breeds through marker-assisted selection, particularly in regions where a black coat color is favored for its market value or adaptive benefits. Incorporating these loci into breeding programs could enhance population consistency and economic performance.

However, due to the complexity of coat color formation and regulation, the specific mechanisms determining animal coat color are not yet fully elucidated. With the advancement of genomic sequencing technologies and molecular testing technology, there is potential for deeper exploration and validation of these mechanisms.

## 5. Conclusions

This study identifies *ASIP*, *AHCY*, and *ITCH* as key coat color candidate genes through a GWAS of Chongqing goats from Southwestern China, which provides novel insights into the genetic basis of coat color variation in goats. By integrating haplotype analysis, gene function annotation, and pathway enrichment, we confirmed the involvement of this gene region in the black coat phenotype of local Chongqing goats and uncovered a potential link between *ITCH* and white coat color expression.

These findings have important implications for local goat populations, particularly in the selective breeding and maintenance of desirable coat color traits in indigenous breeds. Understanding the genetic basis of pigmentation can aid in maintaining the genetic diversity of local goat populations and optimizing breeding strategies for improved economic value.

## Figures and Tables

**Figure 1 animals-15-01849-f001:**
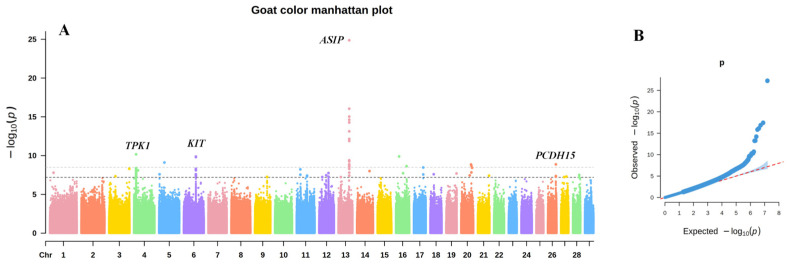
GWAS results for black, white, and black-white coat colors of Chongqing local goats. Note: (**A**) is a Manhattan plot, the gray horizontal line is the very significant threshold line (0.05/n), and the black is the significant threshold line (1/n), the different colors of dots represent SNPs on different chromosomes; (**B**) is a QQ plot, with the red dotted line indicating the expected distribution under the null hypothesis and the blue dots showing the observed *p*-values.

**Figure 2 animals-15-01849-f002:**
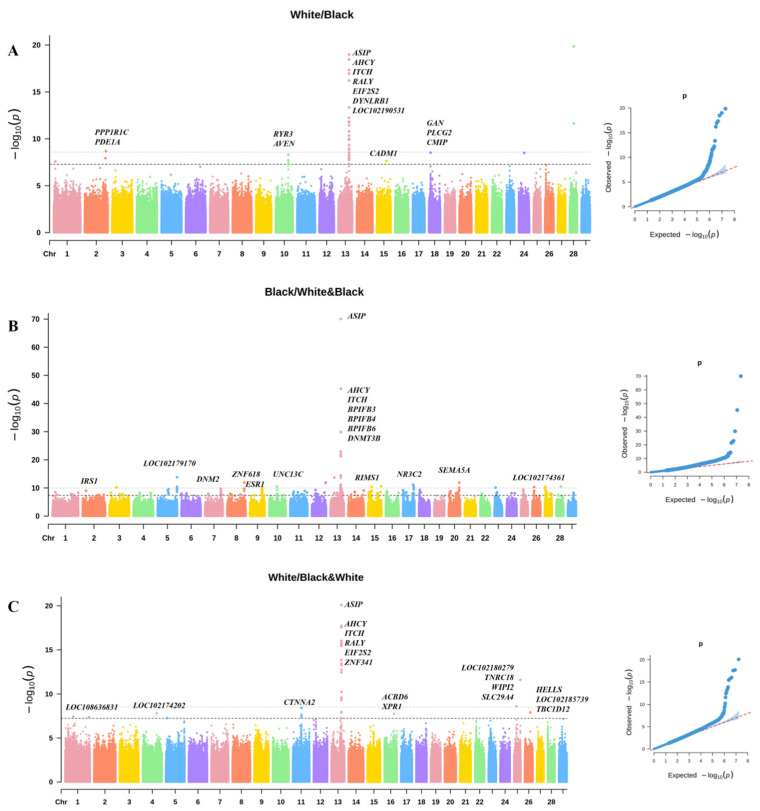
GWAS results for different combinations of coat color. Note: (**A**) is the Manhattan plot and QQ plot of the black and white coat colors; the gray horizontal line is the very significant threshold line (0.05/n) and the black line is the significant threshold line (1/n). (**B**) is the Manhattan plot and QQ plot of the black and black-white coat colors. (**C**) is the Manhattan plot and QQ plot of the white and black-white coat colors. In QQ plots, the red dotted line represents the expected distribution under the null hypothesis, and blue dots represent observed *p*-values.

**Figure 3 animals-15-01849-f003:**
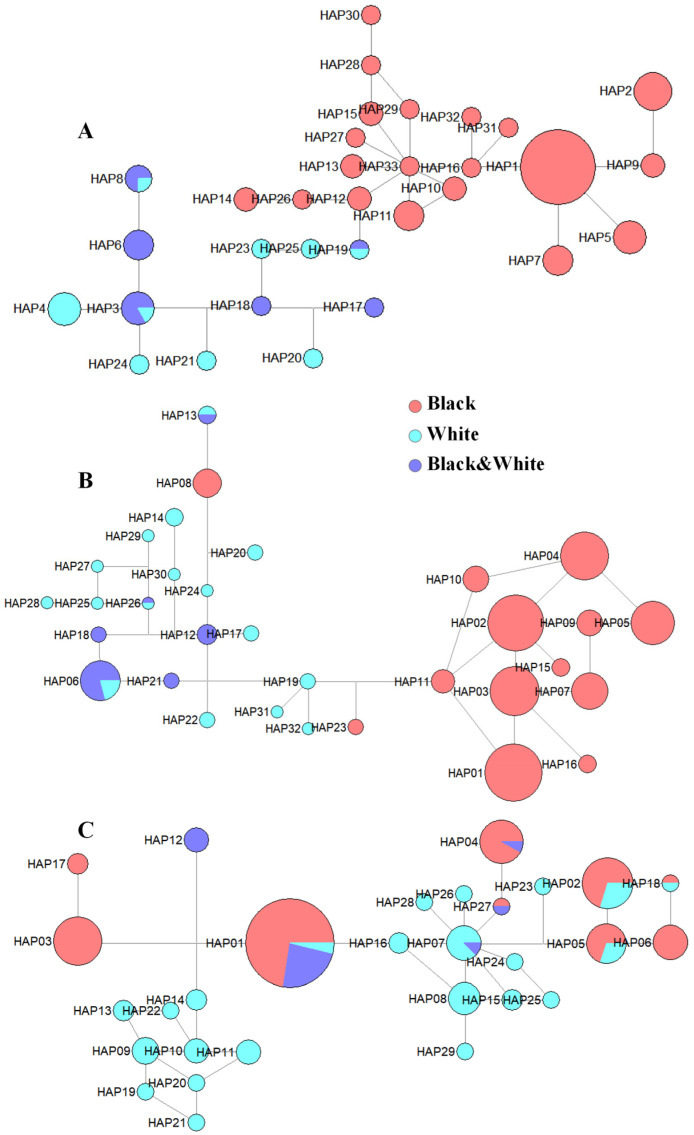
Haplotype analysis results for the significant genes related to coat color in Chongqing local goats. Note: (**A**) represents the haplotype network of the *ASIP* gene across different coat colors; (**B**) represents the haplotype network of the *AHCY* gene across different coat colors; (**C**) represents the haplotype network of the *ITCH* gene across different coat colors. The red color in the network represents the group with a black coat color; the purple color represents the group with a white coat color; and the blue color represents the group with a black-white coat color.

**Figure 4 animals-15-01849-f004:**
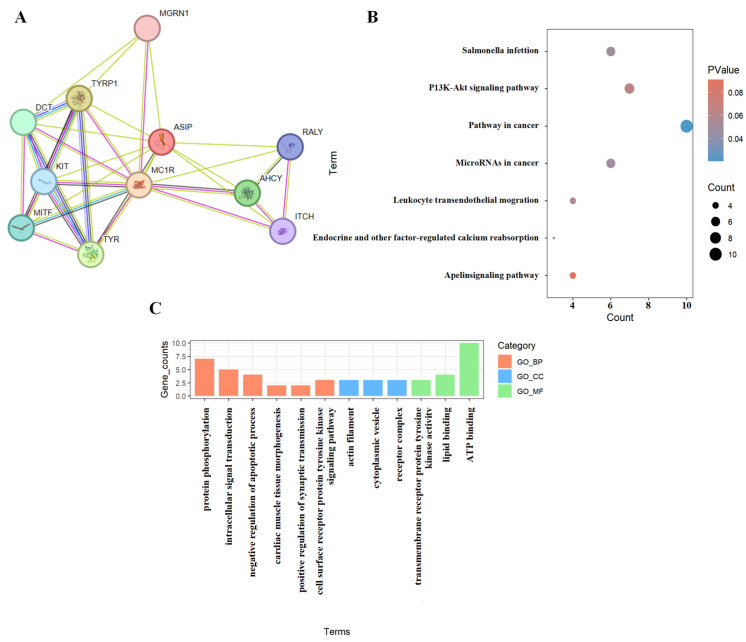
PPI (**A**), KEGG (**B**), and GO (**C**) results for significant genes (*p* < 1/n).

## Data Availability

Data sharing is not applicable to this article as no datasets were generated or analyzed during the current study.

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
