# Peer review of "ASIP, AHCY and ITCH Genes Are Associated with the Coat Color of Local Goats (Capra hircus) of Southwestern China"

_animals, 2025, doi:10.3390/ani15131849_

Round 1
Reviewer 1 Report
Comments and Suggestions for Authors
The job realized was really good. the quality of dates is good. Only my suggestion is to improve a little the relation of data obtained with a better understand of teh application in animal production
Author Response
Dear reviewer,
We would like to thank you for taking the time and effort necessary to review our manuscript. We sincerely appreciate all valuable comments and suggestions, which helped us to improve the quality of the manuscript. The response are as follows.
The job realized was really good. the quality of dates is good. Only my suggestion is to improve alittle the relation of data obtained with a better understand of teh application in animal production.
Response: We sincerely appreciate your positive feedback on the quality of our work and data. Thank you as well for your valuable suggestion regarding the connection between our findings and their application in animal production. In response, we have added the following sentence at the end of the Discussion section to highlight the potential implications for breeding programs: “These findings may be applied to the genetic improvement of local breeds through marker-assisted selection, particularly in regions where black coat color is favored for its market value or adaptive benefits. Incorporating these loci into breeding programs could enhance population consistency and economic performance.” This addition underscores how the coat color-associated loci identified in our study could support marker-assisted selection strategies, particularly in production systems where black pigmentation is economically or environmentally advantageous. We hope this revision enhances the practical relevance of our work.
We hope the revision meet your expectations and look forward to your further comments.
Best regards,
Yongju Zhao and Naiyi Xu
Reviewer 2 Report
Comments and Suggestions for Authors
The study “ASIP, ITCH and AHCY genes are associated with the coat color of local goats (Capra hircus) of southwestern China” is well-designed, methodologically sound, and presents novel findings that contribute significantly to our understanding of pigmentation genetics in livestock. The integration of GWAS, haplotype analysis, and functional enrichment greatly strengthens the validity of the conclusions. The study focuses on a unique and underrepresented population Chongqing local goats providing valuable data for regional breed conservation and improvement. The sample size is commendable, increasing the power and robustness of the GWAS results. The identification of ITCH as a novel candidate gene for white coat color adds a noteworthy dimension to the current literature.
Suggestions:
- The manuscript is generally well-written, but a light language editing round is recommended to improve overall fluency, particularly to address minor issues such as missing articles, inconsistent verb tenses, and occasional awkward phrasing.
- Including breed names directly on the figure S1 or in its caption would improve readability for non-specialist readers.In some figures (e.g., haplotype networks), the color codes representing each group are not clearly defined in the legend. Please consider adding explicit labels for clarity.
- Some terminology, such as “Manhattan plot,” is inconsistently spelled (“Mahattan” in one instance). Please ensure consistency throughout the manuscript.
- On lines 156-159, the sentence "In addition, significant regions on chromosomes 1, 4, 11, 16, 25, and 26, containing genes such as CTNNA2, SLC29A4, and LOC102185739, were also detected and may be associated with coat color variation in goat (Figure 2C, Table S5.)" contains a punctuation error. The period should be placed outside the parentheses to correct the sentence structure.
- It is suggested to change "genes" in the title to "ASIP, ITCH, and AHCY genes" to meet the specification of parallel listing in English.
- Some references are relatively dated. Including more recent studies (within the past 3 years) on goat pigmentation genetics could further highlight the relevance of this work.
Author Response
Dear reviewer,
We would like to thank you for taking the time and effort necessary to review our manuscript. We sincerely appreciate all valuable comments and suggestions, which helped us to improve the quality of the manuscript. The response are as follows.
The study “ASIP, ITCH and AHCY genes are associated with the coat color of local goats (Capra hircus) of southwestern China” is well-designed, methodologically sound, and presents novel findings that contribute significantly to our understanding of pigmentation genetics in livestock. The integration of GWAS, haplotype analysis, and functional enrichment greatly strengthens the validity of the conclusions. The study focuses on a unique and underrepresented population Chongqing local goats providing valuable data for regional breed conservation and improvement. The sample size is commendable, increasing the power and robustness of the GWAS results. The identification of ITCH as a novel candidate gene for white coat color adds a noteworthy dimension to the current literature.
Response: We sincerely thank the reviewer for the positive evaluation and encouraging remarks on our study design, methodology, and scientific contributions.
Suggestions:
- The manuscript is generally well-written, but a light language editing round is recommended to improve overall fluency, particularly to address minor issues such as missing articles, inconsistent verb tenses, and occasional awkward phrasing.
Response: Thank you for your valuable suggestion. We have thoroughly revised the entire manuscript to improve overall language quality. This includes correcting missing articles, aligning verb tenses consistently, and refining sentence structure to eliminate awkward phrasing.
- Including breed names directly on the figure S1 or in its caption would improve readability for non-specialist readers.In some figures (e.g., haplotype networks), the color codes representing each group are not clearly defined in the legend. Please consider adding explicit labels for clarity.
Response: We appreciate your insightful recommendation. In response, we have added breed names Figure S1 in its caption. Additionally, we revised the legends of all relevant figures (e.g., haplotype networks) to clearly define the color codes used for each group。
- Some terminology, such as “Manhattan plot,” is inconsistently spelled (“Mahattan” in one instance). Please ensure consistency throughout the manuscript.
Response: Thank you for pointing this out. We have carefully reviewed the manuscript and corrected the misspelled instance of “Mahattan” to “Manhattan” to ensure consistent and correct terminology usage throughout the manuscript.
- On lines 156-159, the sentence "In addition, significant regions on chromosomes 1, 4, 11, 16, 25, and 26, containing genes such as CTNNA2, SLC29A4, and LOC102185739, were also detected and may be associated with coat color variation in goat (Figure 2C, Table S5.)" contains a punctuation error. The period should be placed outside the parentheses to correct the sentence structure.
Response: We thank the reviewer for catching this punctuation issue. The period has been correctly moved outside the parentheses in the revised manuscript to align with standard punctuation conventions.
- It is suggested to change "genes" in the title to "ASIP,ITCH, and AHCY genes" to meet the specification of parallel listing in English.
Response: Thank you very much for your valuable suggestions regarding the use of English language and gene name formatting. We have carefully reviewed the entire manuscript to improve grammatical consistency and fluency. In addition, we have standardized the terminology throughout the text, including the gene names in the title and main text, to ensure accurate and parallel representation.
- Some references are relatively dated. Including more recent studies (within the past 3 years) on goat pigmentation genetics could further highlight the relevance of this work.
Response: Thank you for this important recommendation. In response, we have updated the reference list to include several relevant and recent studies (published within the past three years) on goat pigmentation genetics. These additions enhance the timeliness and contextual relevance of our work and are cited in both the Introduction and Discussion sections. All changes are highlighted in the revised manuscript.
We hope the revision meet your expectations and look forward to your further comments.
Best regards,
Yongju Zhao and Naiyi Xu
Reviewer 3 Report
Comments and Suggestions for Authors
Dear Authors,
Please find the attached file.
Best regards,
Reviewer

Author Response
Dear reviewer,
We would like to thank you for taking the time and effort necessary to review our manuscript. We sincerely appreciate all valuable comments and suggestions, which helped us to improve the quality of the manuscript. The response are as follows.
The reviewed work was prepared very carefully, using a very large amount of material. Was the consent of the ethics committee required to conduct the research?
Response: Thank you for your kind words and for recognizing of our work. Regarding your question, this study did not involve any direct animal experimentation or sampling. All data used in our analysis were obtained from previously published or publicly available datasets (“Zhang, L.; Duan, Y.; Zhao, S.; Yu, H.; Zhang, J.; Xu, N.; Zhao, Y. Genome-wide association studies reveal novel loci associated with carcass and body measures in goats (Capra hircus). bioRxiv 2025, 2025.2003.2024.644862, doi:10.1101/2025.03.24.644862.”).
The authors presented the problem under study in a very concise manner, successfully demonstrating potential candidate genes for possible further molecular studies. The use of bioinformatics tools (e.g., PPI) explained and validated the participation of the indicated genes in the processes related to the synthesis of pigments by melanocytes.
Response: We sincerely appreciate your positive evaluation of the research rationale and the integration of bioinformatics tools such as PPI to support gene function interpretation. Your recognition encourages us to continue exploring these candidate genes through functional assays in future studies.
Literature item number 8 comes from an unpublished source; if a scientific article has been published in the meantime, please correct it.
Response: Thank you for your helpful comment. The referenced work (item 8) has not yet been formally published in a peer-reviewed journal, but it is publicly available as a preprint on bioRxiv.
Are you sure the address provided for the DAVID tool is correct? Maybe this was the address? https://davidbioinformatics.nih.gov/
Response: We appreciate your attention to this detail. The correct URL for the DAVID tool is: https://david.ncifcrf.gov/ or https://davidbioinformatics.nih.gov/. We have corrected the web address in the revised manuscript to ensure accuracy and accessibility for readers.
Lines 125 and 126 – this is a description for the above graphic, so it should be formatted appropriately (smaller font) and merged. I suggest that after the period in the above description ...goats. continue and add the content from lines 125 and 126. In the typescript it is unnecessarily separated.
Figure 2 description – may consider merging as with figure 1 description.
Figure 3 description – may consider merging as with figure 2 description.
Response: Thank you for your careful reading and constructive suggestions. We have revised the figure captions for Figures 1, 2, and 3 by merging the previously separated descriptive content into each figure legend. The font size and formatting have also been adjusted to match standard figure caption styles.
In the indicated genes (e.g., ASIP) the existence of several important SNPs was demonstrated; did the authors try to verify their functionality? Where were these SNPs located (exons, introns, promoter?)
Response: Thank you for raising this important point. Based on our annotation using SnpEff, the significant SNPs identified within the ASIP region were primarily located in intronic and upstream regulatory regions, suggesting a potential role in transcriptional regulation rather than direct protein-coding alterations. We have added a corresponding sentence in the Discussion to clarify this point and to acknowledge the need for future functional validation studies. In addition, we have summarized the annotation results of significant SNPs in Supplementary Table S6 to provide a clearer overview of their genomic positions and predicted functional categories.
We hope the revision meet your expectations and look forward to your further comments.
Best regards,
Yongju Zhao and Naiyi Xu